# Uncertainty-Based Adaptive Learning for Reading Comprehension

## Abstract

Recent years have witnessed a surge of successful applications of machine reading comprehension. Of central importance to the tasks is the availability of massive amount of labeled data, which facilitates the training of large-scale neural networks. However, in many real-world problems, annotated data are expensive to gather not only because of time cost and budget, but also of certain domain-specific restrictions such as privacy for healthcare data. In this regard, we propose an uncertainty-based adaptive learning algorithm for reading comprehension, which interleaves data annotation and model updating to mitigate the demand of labeling. Our key techniques are two-fold: 1) an unsupervised uncertainty-based sampling scheme that queries the labels of the most informative instances with respect to the currently learned model; and 2) an adaptive loss minimization paradigm that simultaneously fits the data and controls the degree of model updating. We demonstrate on the benchmark dataset that 25% less labeled samples suffice to guarantee similar, or even improved performance. Our results demonstrate a strong evidence that for label-demanding scenarios, the proposed approach offers a practical guide on data collection and model training.

## 1 Introduction

The goal of machine reading comprehension (MRC) is to train an AI model which is able to understand natural language text (e.g. a passage), and answer questions related to it (Hirschman et al., 1999); see Figure 1 for an example. MRC has been one of the most important problems in natural language processing thanks to its various successful applications, such as smooth-talking AI speaker assistants – a technology that was highlighted as among 10 breakthrough technologies by MIT Technology Review very recently (Karen, 2019).

Of central importance to MRC is the availability of benchmarking question-answering datasets, where a larger dataset often enables training of a more informative neural networks. In this regard, there have been a number of benchmark datasets proposed in recent years, with the efforts of pushing forward the development of MRC. A partial list includes SQuAD (Rajpurkar et al., 2016), NewsQA (Trischler et al., 2017), MSMARCO (Nguyen et al., 2016), and Natural Questions (Kwiatkowski et al., 2019). While the emergence of these high-quality datasets have stimulated a surge of research and have witness a large volume of deployments of MRC, it is often challenging to go beyond the scale of the current architectures of neural networks, in that it is extremely expensive to obtain massive amount of labeled data. The barrier of data collection can be seen from SQuAD: the research group at Standford University spent 1,547 working hours for the annotation of SQuAD dataset, with the cost over $14,000. This issue was also set out and addressed by AI companies. However, even equipped with machine learning assisted labeling tools (e.g. Amazon SageMaker Ground Truth), it is still expensive to hire and educate expert workers for annotation. What makes the issue more serious is that there is a rise of security and privacy concerns in various problems, which prevents researchers from scaling their projects to diverse domains efficiently. For example, all annotators are advised to get a series of training about privacy rules, such as Health Insurance Portability & Accountability Act, before they can work on the medical records.

In this work, we tackle the challenge by proposing a computationally efficient learning algorithm that is amenable for label-demanding problems. Unlike prior MRC methods that separate data annotation and model training, our algorithm interleaves these two phases. Our algorithm, in spirit,

> - **Question**: What causes precipitation to fall?
> - **Passage**: In meteorology, precipitation is any product of the condensation of atmospheric water vapor that falls under `gravity`. The main forms ... intense periods of rain in scattered locations are called "shower".
> - **Answer**: `gravity`

Figure 1: An illustrative example in the SQuAD dataset (Rajpurkar et al., 2016).

resembles the theme of active learning (Balcan et al., 2007), where the promise of active learning is that we can always concentrate on fitting only the *most informative examples* without suffering a degraded performance. While there have been a considerable number of works showing that active learning often guarantees exponential savings of labels, the analysis holds typically for linear classification models Awasthi et al. (2017); Zhang (2018); Zhang et al. (2020). In stark contrast, less is explored for the more practical neural network based models since it is nontrivial to extend important concepts such as large margin of linear classifiers to neural networks. As a remedy, we consider an unsupervised sampling scheme based on the uncertainty of the instances (Settles, 2009). Our sampling scheme is adaptive (i.e. active) in the sense that it chooses instances that the currently learned model is most uncertain on. To this end, we recall that the purpose of MRC is to take as input a passage and a question, and finds the most accurate answer from the passage. Roughly speaking, this can be thought of as a weight assignment problem, where we need to calculate how likely each word span in the passage could be the correct answer. Ideally, we would hope that the algorithm assigns 1 to the correct answer, and assigns 0 to the remaining, leading to a large separation between the correct and those incorrect. Alternatively, if the algorithm assigns, say 0.5 to two different answers and assigns 0 to others, then it is very uncertain about its response – this is a strong criterion that we need to query the correct answer to an expert, i.e. performing active labeling. Our uncertainty-based sampling scheme is essentially motivated by this observation: the uncertainty of an instance (i.e. a pair of passage and question) is defined as the gap between the weight of the best candidate answer and the second best. We will present a more formal description in Section 2.

After identifying these most uncertain, and hence most informative instances, we query their labels and use them to update the model. In this phase, in addition to minimize the widely used entropy-based loss function, we consider an adaptive regularizer which has two important properties. First, it enforces that the new model will not deviate far from the current model, since 1) with reasonable initialization we would expect that the initial model should perform not too bad; and 2) we do not want to overfit the data even if they are recognized as informative. Second, the regularizer has a coefficient that is increasing with iterations. Namely, as the algorithm proceeds the stability of model updating outweighs loss minimization. In Section 2 we elaborate on the concrete form of our objective function. It is also worth mentioning that since in each iteration, the algorithm only fits the uncertain instances, the model updating is more faster than traditional methods.

The pipeline is illustrated in Figure 2. Given abundant unlabeled instances, our algorithm first evaluates their uncertainty and detects the most informative ones, marked as red. Then we send these instances to an expert to obtain the groundtruth answers, marked as yellow. With the newly added labeled samples, it is possible to perform incremental updating of the MRC model.

**Roadmap.** We summarize our main technical contributions below, and discuss more related works in Section 5. In Section 2 we present a detailed description of the core components of our algorithm, and in Section 3 we provide an end-to-end learning paradigm for MRC with implementation details. In Section 4, we demonstrate the efficacy of our algorithm in terms of exact match, F-1 score, and the savings of labels. Finally we conclude this paper in Section 6.

## 1.1 SUMMARY OF CONTRIBUTIONS

We consider the problem of learning an MRC model in the label-demanding context, and we propose a novel algorithm that interleaves data annotation and model updating. In particular, there are two core components for this end: an unsupervised uncertainty-based sampling scheme that only queries labels of the most informative instances with respect to the currently learned model, and

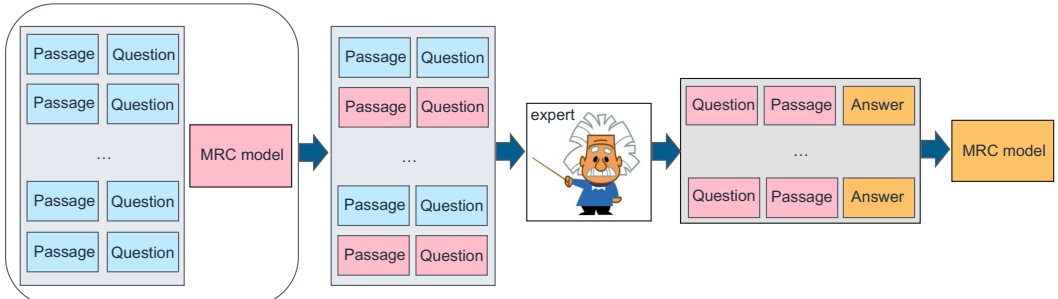

Figure 2: **Illustration of our learning algorithm in each iteration.** Given a pool of unlabeled pairs of passage-questions, the algorithm first identifies the instances that it is most uncertain on, e.g. those marked in red. Then it queries to an expert to gather the answers (i.e. labels), and restricts to fit the newly labeled instances.

an adaptive loss minimization paradigm that simultaneously fits the data and controls the degree of model updating. Moreover, our approach is modular in nature, meaning that the community would benefit from this work by leveraging our techniques into more real-world problems (e.g. image classification) where the availability of labels is a major concern.

## 2 ALGORITHM

In this section, we formally introduce the problem setup and our main algorithm ALBUS (Algorithm 1). We use $x := (p, q)$ to represent a pair of passage $p$ and question $q$, which is also called an unlabeled instance, or simply an instance. If there are multiple questions, say $q_1, q_2$, to a same passage $p$, we will use two instances $x_1 := (p, q_1)$ and $x_2 := (p, q_2)$. Given an instance $x$, our goal is to predict an answer. We use a zero-one vector $a$ to indicate the correct answer, and $(x, a)$ is called a labeled instance. The prediction made by the learner is denoted by $\hat{a}$. We will always assume that all the coordinates of $\hat{a}$ are non-negative, and their sum equals one, which can be easily satisfied if the last layer of the neural network is softmax.

### 2.1 UNSUPERVISED UNCERTAINTY-BASED RANDOM SAMPLING

Since data annotation is expensive, we treat the problem as such that all the instances are unlabeled before running the algorithm, and as the algorithm proceeds, it may adaptively detects the most informative instances to be labeled by experts or crowd workers. Thus, the central questions to learning are: 1) how to measure the informativeness of the unlabeled instances in a computationally efficient manner; and 2) how to select a manageable number of instances for annotation (since the algorithm might identify bunch of useful instances). We address both questions in the following.

#### 2.1.1 METRIC OF INFORMATIVENESS

**Intuition.** We first address the first question, i.e. design a metric to evaluate the informativeness. To ease the discussion, suppose that for a given instance $x$, there are only two answers to choose from, i.e. $a$ is a two-dimensional vector, and that the algorithm has been initialized, e.g. via pre-training. If the current model takes as input $x$, and predicts $\hat{a} = (1, 0)$, then we think of this instance as less informative, in that the algorithm has an extremely high confidence on its prediction.[1] On the other spectrum, if the prediction $\hat{a} = (0.5, 0.5)$, then it indicates that the current model is not able to distinguish the two answers. Thus, sending the correct answer $a$ together with the instance to the algorithm will lead to significant progress.

We observe that underlying the intuition is a notion of separation between the answer with highest confidence and that with second highest, denoted by $\Delta_w(x)$, where $w$ denotes the current model

---

[1]The algorithm may of course make a mistake, but this will be treated by future model updating. Here we are just giving an intuitive explanation following the idealized scenario.

---

**Algorithm 1** ALBUS: Adaptive Learning By Uncertainty-Based Sampling

---

**Require:** a set of unlabeled instances $U = \{\boldsymbol{x}_1, \ldots, \boldsymbol{x}_n\}$, initial MRC model $\boldsymbol{w}_0$, maximum iteration number $T$, thresholds $\{\tau_1, \ldots, \tau_T\}$, number of instances to be labeled $n_0$.
**Ensure:** A new MRC model $\boldsymbol{w}_T$.
1: $U_1 \leftarrow U$.
2: **for** $t = 1, \cdots, T$ **do**
3:     Compute $\Delta_{\boldsymbol{w}_{t-1}}(\boldsymbol{x})$ for all $\boldsymbol{x} \in U_t$.
4:     $B_t \leftarrow \{\boldsymbol{x} \in U_t : \Delta_{\boldsymbol{w}_{t-1}}(\boldsymbol{x}) \leq \tau_t\}$.
5:     Compute the sampling probability $\Pr(\boldsymbol{x})$ for all $\boldsymbol{x} \in B_t$.
6:     $S_t \leftarrow$ randomly choose $n_0$ instances from $B_t$ by the distribution $\{\Pr(\boldsymbol{x})\}_{\boldsymbol{x} \in B_t}$, and query their labels.
7:     Update the model $\boldsymbol{w}_t \leftarrow \arg\min_{\boldsymbol{w}} L(\boldsymbol{w}; S_t)$.
8:     $U_{t+1} \leftarrow U_t \backslash S_t$.
9: **end for**

---

parameters. In fact, let our algorithm be a function $f_{\boldsymbol{w}} : \boldsymbol{x} \mapsto \hat{\boldsymbol{a}}$. Denote by $\hat{a}^{(1)}$ and $\hat{a}^{(2)}$ the highest and second highest value in $\hat{\boldsymbol{a}}$. Then

$$\Delta_{\boldsymbol{w}}(\boldsymbol{x}) = \hat{a}^{(1)} - \hat{a}^{(2)}. \tag{1}$$

Given the unlabeled training set $\{\boldsymbol{x}_1, \boldsymbol{x}_2, \ldots, \boldsymbol{x}_n\}$ and the currently learned model, we can evaluate the degree of separation $\{\Delta_1, \Delta_2, \ldots, \Delta_n\}$ where we write $\Delta_i := \Delta_{\boldsymbol{w}}(\boldsymbol{x}_i)$ to reduce notation clutter since most of the time, the model $\boldsymbol{w}$ is clear from the context. This answers the first question proposed at the beginning of the section, i.e. how to measure the informativeness of the instances.

### 2.1.2 UNCERTAINTY-BASED SAMPLING

It remains to design a mechanism so that we can gather a manageable number of instances to be labeled. A natural approach will be specifying the maximum number $n_0$, so that in each iteration the algorithm chooses at most $n_0$ instances with lowest degree of separation. Yet, we observe that for some marginal cases, many instances have very close $\Delta_i$, e.g. $\Delta_1 = 0.101$ and $\Delta_2 = 0.102$. Using the above strategy may annotate $\boldsymbol{x}_1$ while throwing away $\boldsymbol{x}_2$. From the practical perspective, however, we hope both instances have a chance to be selected to increase diversity. Henceforth, we consider a "soft" approach based on random sampling in this paper.

Fix an iteration $t$ of the algorithm. First, we define a threshold $\tau_t \in (0, 1]$. Based on the current model $\boldsymbol{w}_{t-1}$, we calculate $\Delta_1, \ldots, \Delta_n$. Then we obtain a sampling region

$$B_t := \{\boldsymbol{x}_i : \Delta_i \leq \tau_t\}, \tag{2}$$

which contains informative instances (recall that a lower degree of separation implies more informative). Inspired by the probability selection scheme (Abe & Long, 1999), we define the sampling probability as

$$\Pr(\boldsymbol{x}) = \begin{cases} \frac{1}{|B_t| + \gamma(\Delta_{\boldsymbol{x}} - \Delta_{\boldsymbol{x}^*})}, & \forall \boldsymbol{x} \in B_t \backslash \boldsymbol{x}^*, \\ 1 - \sum_{\boldsymbol{x}' \neq \boldsymbol{x}^*} \frac{1}{|B_t| + \gamma(\Delta_{\boldsymbol{x}'} - \Delta_{\boldsymbol{x}^*})}, & \text{when } \boldsymbol{x} = \boldsymbol{x}^*. \end{cases} \tag{3}$$

In the above expression, $\boldsymbol{x}^*$ is the instance in $B_t$ with lowest degree of separation, i.e. the most uncertain instance; $\gamma \geq 0$ is a tunable hyper-parameters. Observe that if $\gamma = 0$, it becomes uniform sampling. In addition, in view of the sampling probability in (3), the instance $\boldsymbol{x} \neq \boldsymbol{x}^*$ will be sampled with probability less than $1/|B_t|$, and $\boldsymbol{x}^*$ is sampled with probability more than $1/\mu$, as

$$\Pr(\boldsymbol{x}^*) \geq 1 - \sum_{\boldsymbol{x}' \neq \boldsymbol{x}^*} \frac{1}{|B_t|} = 1 - \frac{|B_1| - 1}{|B_t|} = \frac{1}{|B_t|}. \tag{4}$$

Therefore, the sampling scheme always guarantees that $\boldsymbol{x}^*$ will be selected with highest probability, and if needed, it is possible to make this probability close to 1 by increasing $\gamma$. In our algorithm, we set $\gamma = \Theta(\sqrt{|B_t|})$ which works well in practice.

## 2.2 Adaptive Loss Minimization

Another crucial component in ALBUS is loss minimization. Here our novelty is an introduction of an adaptive regularizer that balances the progress of model updating and per-iteration data fitting.

Let $S_t$ be the set of labeled instances determined by our random sampling method at the $t$-th iteration. For any $(\boldsymbol{x}, \boldsymbol{a}) \in S_t$, since $\boldsymbol{a}$ is an indicator vector, the problem can be thought of as multi-classification. Therefore, a typical choice of sample-wise loss function is logistic loss, denoted by $\ell(w; \boldsymbol{x}, \boldsymbol{a})$, which can be easily implemented by using a softmax layer in the neural network. On top of the logistic loss, we also consider an adaptive $\ell_2$-norm regularizer, which gives the following objective function:

$$L(\boldsymbol{w}; S_t) := \frac{1}{|S_t|} \sum_{(\boldsymbol{x}, \boldsymbol{a}) \in S_t} \ell(\boldsymbol{w}; \boldsymbol{x}, \boldsymbol{a}) + \frac{\lambda_t}{2} \|\boldsymbol{w} - \boldsymbol{w}_{t-1}\|^2 . \tag{5}$$

Different from the broadly utilized $\ell_2$-norm regularizer $\|\boldsymbol{w}\|^2$, we appeal to a *localized* form, in the sense that the objective function pushes the updated model to be close to the current model $\boldsymbol{w}_{t-1}$ under Euclidean distance. This is motivated by the fact that in many cases, warm starting the algorithm with pre-training often exhibits favorable performance. Hence, though we want the model to be adapted to the new dataset, we carefully control the progress of model updating so that it does not deviate far from the current.

Regarding the coefficient $\lambda_t$, we increase it by a constant factor greater than one in each iteration. Therefore, as the algorithm proceeds, the localization property plays a more important role than the logistic loss. Our treatment is inspired by the literature of active learning, where similar localized $\ell_2$-norm constraint is imposed (Balcan et al., 2007; Zhang et al., 2020). This can be viewed as a stability property of our algorithm, and we discover that it works very well on benchmark datasets.

## 3 Implementation Details

**Uncertainty-based sampling.** We introduce how to select the batch $S_t$ in each iteration with current MRC model $\boldsymbol{w}_{t-1}$. For a given pair of $(\boldsymbol{p}, \boldsymbol{q})$, an answer is of the form of a word span from the $i$-th position to the $j$-th position of the passage. Given the span $(i, j)$ and the passage $\boldsymbol{p}$, we use BERT (Devlin et al., 2019) as our embedding method, which produces a feature description denoted by $E_{\boldsymbol{p}}(i, j)$. We then construct a probability matrix $\hat{M}$ whose $(i, j)$-th entry $\hat{M}_{i,j}$ is given by the following:

$$\hat{M}_{i,j} = \frac{\exp(\boldsymbol{w}_{t-1} \cdot E_{\boldsymbol{p}}(i, j))}{\sum_{i', j'} \exp(\boldsymbol{w}_{t-1} \cdot E_{\boldsymbol{p}}(i', j'))}. \tag{6}$$

Observe that the matrix $\hat{M}$ forms a distribution over all possible word spans, i.e. all possible answers. It is then straightforward to convert $\hat{M}$ into the vector $\hat{\boldsymbol{a}}$, for example, by concatenating all the columns. Based on the obtained answer $\hat{\boldsymbol{a}}$, we are able to perform uncertainty-based sampling as discussed in Section 2.

**Adaptive loss minimization.** We already derived the probability matrix $\hat{M}$ in (6). During loss minimization, i.e. supervised fine-tuning, we aim to update $\boldsymbol{w}_{t-1}$ by minimizing $L(w; S_t)$. Since we have clarified the regularizer, it suffices to give the detailed form of the loss $\ell(\boldsymbol{w}; \boldsymbol{x}, \boldsymbol{a})$ where we recall that $\boldsymbol{x} = (\boldsymbol{p}, \boldsymbol{q})$. Note that using the groundtruth answer $\boldsymbol{a}$, we know the correct span $(i_{\boldsymbol{a}}, j_{\boldsymbol{a}})$ for question $\boldsymbol{q}$. Thus, the likelihood that we observe $S_t$ is

$$\Pr(S_t) = \prod_{(\boldsymbol{p}, \boldsymbol{q}, \boldsymbol{a}) \in S_t} \frac{\exp(\boldsymbol{w} \cdot E_{\boldsymbol{p}}(i_{\boldsymbol{a}}, j_{\boldsymbol{a}}))}{\sum_{i', j'} \exp(\boldsymbol{w} \cdot E_{\boldsymbol{p}}(i', j'))} \tag{7}$$

The loss function $\ell(\boldsymbol{w}; S_t)$ is simply the negative log-likelihood.

## 4 Experiments

**Datasets.** We focus on the span-based datasets, namely Stanford Question Answering Dataset (SQuAD) (Rajpurkar et al., 2016) and NewsQA (Trischler et al., 2017).SQuAD consists of over

Table 1: EM and F1 score on the SQuAD dataset.

| #Labels queried | EM | | | | | | F1 score | | | | | |
|---|---|---|---|---|---|---|---|---|---|---|---|---|
| | Badge | Conf | Entropy | Margin | Rand | Ours | Badge | Conf | Entropy | Margin | Rand | Ours |
| 5000 | 60.94 | 59.62 | 60.52 | 62.71 | 62.58 | **64.03** | 72.20 | 72.29 | 72.78 | 74.28 | 73.97 | **75.30** |
| 15000 | 71.75 | 72.05 | 71.89 | 72.69 | 71.54 | **74.13** | 81.62 | 82.18 | 82.31 | 82.59 | 81.41 | **83.50** |
| 21000 | 73.88 | 74.38 | 74.67 | 74.31 | 73.75 | **75.48** | 83.54 | 83.90 | 84.23 | 83.77 | 83.08 | **84.53** |
| 41000 | 77.55 | 77.37 | 77.80 | 78.16 | 75.86 | **79.02** | 86.02 | 85.69 | 85.95 | 86.19 | 84.51 | **87.09** |
| 61000 | 77.90 | 77.98 | 77.75 | 78.06 | 77.98 | **80.44** | 86.15 | 85.86 | 85.43 | 86.32 | 86.10 | **88.13** |
| 81000 | 76.08 | 76.08 | 75.66 | 76.34 | 78.63 | **81.14** | 84.75 | 84.08 | 83.74 | 84.64 | 86.98 | **88.53** |

100k questions posed by crowdworkers on a set of 536 Wikipedia articles. We use the original split 87,599 questions for training and test on the 10,570 questions. NewsQA is a machine comprehension dataset of over 100k human-generated question-answer pairs from over 10k news articles from CNN. The dataset consists of 74,160 questions for training and 4,212 questions for validation [2].

**Evaluation Metrics.** We use two metrics: Exact Match (EM) and F1 score. EM measures the percentage of predictions that matches any one of the annotated answers exactly. EM gives credit for predictions that exactly match (one of) the gold answers. F1 score measures the average overlap between the prediction and the annotated answer.

**Baselines.** We compare against the following baseline algorithms:

- Badge (batched based sampling) (Ash et al., 2020): it learns the gradient embedding of samples and selects a set of samples by $k$-MEANS++ (Arthur & Vassilvitskii, 2007).

- Conf (confidence sampling) (Wang & Shang, 2014): it is an uncertainty-based algorithm that selects samples with lowest class probability.

- Entropy (Wang & Shang, 2014): it selects samples based on the entropy of the predicted probability distribution.

- Marg (margin-based sampling) (Roth & Small, 2006): it also checks the degree of separation as our algorithm, but selects the $n_0$ lowest rather than performing random sampling as we did.

- Rand (Random sampling): It is the naive baseline of uniformly randomly selecting samples from unlabeled set.

**Other Settings.** To ensure a comprehensive comparison among state-of-the-art approaches, we simulate the annotation process with human experts in the loop by selecting a fixed number of examples $n_0$ to query their labels from training set in each iteration (we set $n_0 = 2,000$ for SQuAD and $n_0 = 5,000$ for NewsQA). The labeled data is used to update the MRC model. We report the exact match and F1 score with the number of iterations. The BERT-base is used as the pretrained model and fine-tuned for 2 epochs with a learning rate of $3e - 5$ and a batch size of 12 [3]. The MRC model is initialized with 1,000 labeled samples for SQuAD and 10,000 for NewsQA. The parameter $\tau_0$ is tuned from the range of $[0.01, 0.1]$ on the training set and decreases at the rate of 1.1.

**Results.** Figure 3 and Figure 4 present EM and F1 score with the increase of the number of labeled samples selected by various active learning algorithms. We show the results with all labeled data (Figure 3(a) and Figure 4(a)) and 20,000 labeled data (Figure 3(b) and Figure 4(b)). Our algorithm outperforms state-of-the-art active learning algorithms in almost all the cases.

Table 1 lists some detailed results with a specific number of labeled samples. Our algorithm reaches the best performance in all cases and the advantage is significant specially with a small subset of labeled samples available. For example, in the case of 5,000 labeled examples, our algorithm reaches the EM of 64.07 % while the best of compared algorithms is 62.71 %. Figure 3 and Figure 4 plot the trend EM and F1 score with the rise of labeled examples on SQuAD dataset. We observe that all active learning algorithms reach the best performance before accessing all labeled data compared with Rand. It demonstrates the active learning effectively reduces the number of required labeled data for learning process. Specifically, our algorithm reaches EM 80.44 % and F1 score 88.53 % with 61,000 queries which is close to the best result but with 25% less labeled samples. We can observe the same advantage of our algorithm on the NewsQA dataset as shown in Figure 5.

---

[2]https://github.com/mrqa/MRQA-Shared-Task-2019
[3]https://github.com/huggingface/transformers/tree/master/examples/question-answering

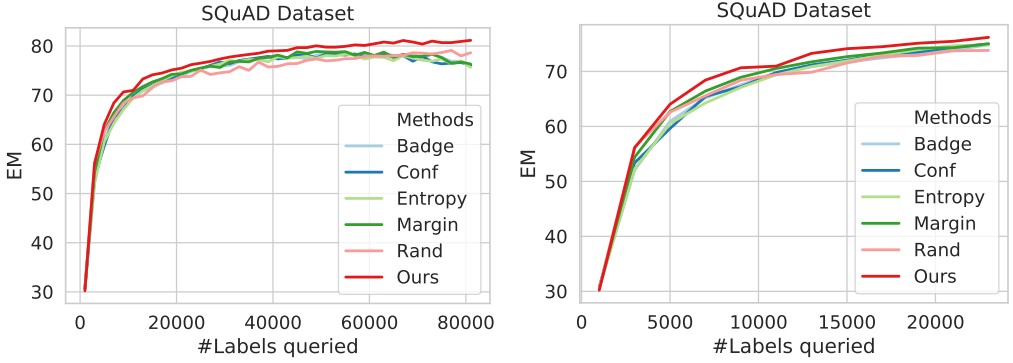

Figure 3: Compared results of EM on SQuAD with over 80k (left) and 20k (right) labeled data.

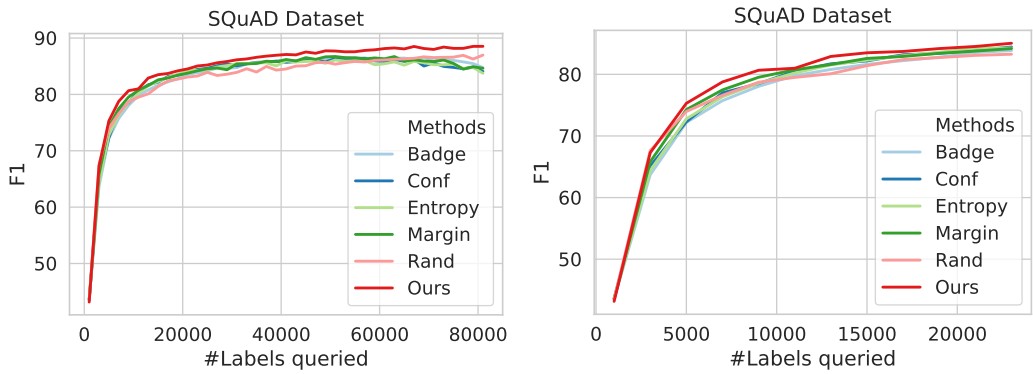

Figure 4: Compared results of F1 score on SQuAD with over 80k (left) and 20k (right) labeled data.

## 5    RELATED WORKS

**Machine Reading Comprehension.** MRC is the ability to read text and answer questions about it. It is a challenging task as it requires the abilities of understanding both the questions and the context. A data-driven approach to reading comprehension goes back to (Hirschman et al., 1999). There are a number of works have proposed to create datasets (Rajpurkar et al., 2016; 2018; Kwiatkowski et al., 2019; Reddy et al., 2019). For example, Stanford Question Answering Dataset (SQuAD) dataset consists of 100K questions on a set of Wikipedia articles (Rajpurkar et al., 2016). Natural Questions dataset (Kwiatkowski et al., 2019) consists of queries issued to the Google search engine, the wikipedia page, long answers and short answers.

Recently, researchers are devoted to develop unsupervised deep learning frameworks to learn the word representation based on a batch of unlabeled data which could be simply fine-tuned for multiple downstream tasks. For example, ELMo (Peters et al., 2018) learned forward and backward language models: the forward one reads the text from left to right, and the other one encodes the text from right to left. GPT (Radford et al., 2018) used a left-to-right Transformer to predict a text sequence word-by-word. Devlin et al. (2019) designed BERT to pre-train deep bidirectional representations from unlabeled text by jointly conditioning on both left and right context in all layers. There are some following works aiming to improve the framework of BERT for different language modeling tasks (Yang et al., 2019; Dai et al., 2019; Dong et al., 2019).BERT significantly improves the performance of natural language understanding tasks. Our work is based on a pretrained BERT model. We fine-tuned BERT with one additional output layer to create the model for the reading comprehension task following (Devlin et al., 2019).

Another direction in reading comprehension is to explore different real-world settings. For example, in open-domain reading comprehension, the passage that contains the answer is not provided but requires retrieval from the knowledge pool (Wang et al., 2019). Yue et al. (2020) considered language

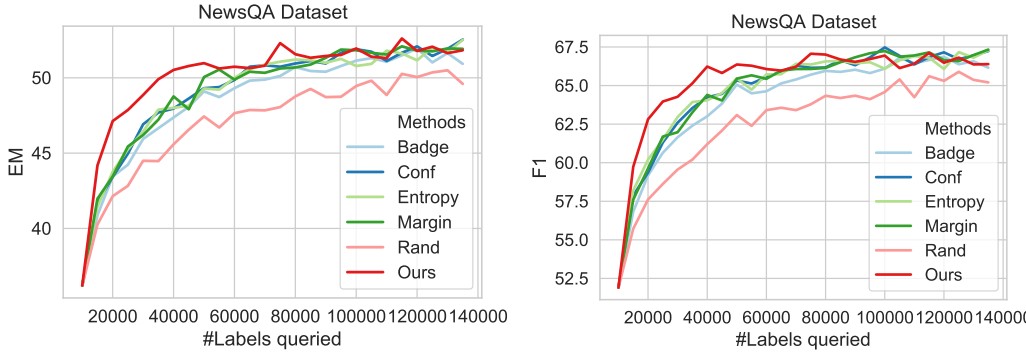

Figure 5: Compared results of EM and F1 score on NewsQA with increase of labeled data.

understanding of clinical data. This work considers the general reading comprehension setting in which the question and related passage are provided.

**Active Learning.** Active learning is a machine learning paradigm that mainly aims at reducing label requirements through interacting with the oracle (experts/annotators) (Settles, 2009).

Active learning has been well studied in both theory and applications. One popular branch of theoretical works follows the PAC active learning setting: researchers are engaged to reduce the label complexity, i.e. the number of query requests, with the error bound guarantee for the produced halfspace with high probability Valiant (1984); Balcan et al. (2009); Hanneke (2014). There are some exciting works focusing on active halfspace learning, such as margin based active learning (Balcan et al., 2007; Balcan & Long, 2013). These approaches learn a classifier from the class of homogeneous linear classifiers, to predict labels from instances.

Existing active learning approaches can be roughly divided to uncertainty-based sampling and representative sampling (Settles, 2009). Representative sampling based approaches select samples that are representative of the whole unlabeled dataset. It can be achieved by performing an optimization minimizing the difference between the selected subset and the global dataset (Sener & Savarese, 2018; Gissin & Shalev-Shwartz, 2019). The uncertainty sampling based algorithms select samples that maximally reduce the uncertainty the algorithm has on a target learning model, such as samples lying closest to the current decision boundary (Tür et al., 2005). The work in this paper belongs to uncertainty-based sampling but using a novel sampling scheme tailored to MRC.

Active learning has shown outstanding performance in real-world applications, such as computer vision (Joshi et al., 2009) and natural language processing (Culotta & McCallum, 2005; Reichart et al., 2008). For example, Shen et al. (2004) combined multiple criteria of active learning for named entity recognition. Recent studies combining deep neural networks and active learning approaches have been proposed (Wang et al., 2016; Zhang et al., 2017; Shen et al., 2018; Geifman & El-Yaniv, 2019). However, these approaches do not consider the correlation between adaptively learned models of selected samples. There are some related work about active learning in visual question answering (Lin & Parikh, 2017). However, little is known about active learning for machine reading comprehension.

## 6 CONCLUSION AND FUTURE WORKS

In this work, we have proposed a novel adaptive learning algorithm for the reading comprehension task. There are two crucial components in our algorithm: an unsupervised uncertainty-based random sampling scheme, and a localized loss minimization paradigm, both of which are adaptive to the currently learned model. We have described the strong motivation of using these techniques, and our empirical study serves as a clear evidence that our algorithm drastically mitigates the demand of labels on large-scale datasets. We highlight that our approach is not essentially tied to MRC, and we expect that it can be extended to other label-demanding problems in natural language processing and image classification.

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
