# OpenReview forum: "Uncertainty-Based Adaptive Learning for Reading Comprehension"
_ICLR.cc/2021/Conference — Reject_

### Official Review · AnonReviewer4 · 2020-10-28
**New uncertainty-based adaptive learning method for passage based question answering but weak evaluation and analysis**

**Rating:** 4
**Confidence:** 4

**Review:**

**Update after author response:** I appreciate the authors' efforts to address my concerns. Thanks for the correction on QBC, I appreciate it. I still believe that the paper needs to accompany a more comprehensive evaluation and qualitative insights to highlight the effectiveness of the proposed method. For instance, it is common practice in Active Learning to report mean accuracy over multiple runs of the same experiment as data is sampled based on a particular heuristic which isn't always deterministic. Furthermore, the choice of warmstart samples could also influence the results, which is why it is recommended to conduct multiple runs of the same experiments and report mean performance. In such a scenario, any claims that arise from only one run of the experiments (as in this paper) should be taken with a grain of salt. I also appreciate the pointer to Equation 4 but how does it translate in practice is another important piece that is missing. BERT based models produce highly confident predictions and to visualize the distribution from your empirical investigation would help bridge the divide between the equations and the empirical results (how they actually turn out in practice). I am also not convinced by the authors' response to why the difference in performance of the BERT model trained on randomly sampled data vs the model trained on data sampled via ALBUS stays within the 1-2% range, often increasing with more data. I believe there are critical questions regarding this paper that need to be addressed before the paper is published, hence, my score remains unchanged.

---------------------------------------------------------------------------------------------------------------------------------------------------------------------------

Summary: The paper proposes a new method to actively sample data to train models in a limited labeled data regime. Focusing on passage based question answering, the authors employ an uncertainty based sampling strategy by selecting examples on which a model has little top-1 prediction confidence. In every training iteration, the model is trained with an adaptive regularizer in an attempt to keep the new model closer to the current model. The results presented in the paper show improvement over three active learning baselines as well as training over iid sampled data on the SQuAD dataset.

Pros:
- The paper looks at an important problem which is little studied in context of deep learning based NLP systems, particularly with pre-trained transformers.
- The two components that together form ALBUS are simple and the authors show that they perform better than various active learning baselines as well as random sampling on the SQuAD dataset.
- I like the approach of ensuring that the model weights do not deviate too much as this also addresses the challenges faced in active learning when we end up sampling outliers.
- The paper is easy to follow.

Cons:
- All evaluations are conducted using a Bert base model. It is unclear whether this method generalizes to other models or not. Furthermore, the benefits of this approach on NewsQA are not easily interpretable, in fact the plots suggest that in many cases, baseline methods outperform this method. What happens when less than 20000 NewsQA examples are sampled for labeling is also not shown so it is hard to meaningfully interpret the results.
- Common and effective Active Learning baselines such as Query by Committee have not been compared against.
- The uncertainty based sampling method samples examples on which a model is less confident (determined by a threshold) on its top-1 prediction, however, the paper presents no qualitative analysis to show that this approach is indeed meaningful in context of models like BERT which, to the best of my knowledge, happen to have very confident predictions. It would help if the authors show how the distribution of the metric of informativeness over examples, and as more and more data is labeled.

Questions for authors for rebuttal:
- It would be great to share actual numbers for NewsQA besides the existing plots. It is clear from the table and the figure that your method consistently outperforms baselines on SQuAD but it is less clear as to what happens in NewsQA as the only figures present are less clear and show that other baselines often outperform your proposed method.
- Several experimental details are missing from the paper that make it hard to put the numbers into context. Are the results averaged over multiple runs? If yes, how many? And if not, then I’d recommend averaging results for each approach over at least 3 runs.
- Since the authors have not shared their code with the submission, it is important to share how certain hyperparameter values were obtained. What influenced the choice to finetune BERT for only 2 epochs for both SQuAD and NewsQA or what influenced the choice of the number of datapoints for warm starting, etc.?
- On SQuAD, it appears that the difference in performance of the BERT model trained on randomly sampled data vs the model trained on data sampled via ALBUS stays within the 1-2% range, sometimes increasing with more data. Intuitively, this gap should be decreasing with more data (which does happen with other baselines that you’ve compared against). Why do you think this gap stays more or less the same with your method regardless of the amount of labeled data?

Typos and other suggestions:
- Page 2, second paragraph last line: “more faster” -> faster
- Page 2, first line: a more appropriate reference for Active Learning would be Cohn et al., 1996. Cohn, D. A., Ghahramani, Z., & Jordan, M. I. (1996). Active learning with statistical models. Journal of artificial intelligence research, 4, 129-145.
- Even though the paper is easy to follow, I believe the writing can be improved, some sections appear to be written in a casual manner, for instance (these are representative, not exhaustive):
(i) line 4 of introduction “smooth-talking AI systems.” Consider using a better term to describe these systems.
(ii) lines 13 and 14 of abstract “demonstrate … that 25% less labeled samples suffice to guarantee.” Since the paper does not establish theoretical guarantees for this method, I would recommend not to use words like guarantee in this context.

Missing references:

- Siddhant, A., & Lipton, Z. C. (2018). Deep Bayesian Active Learning for Natural Language Processing: Results of a Large-Scale Empirical Study. In Proceedings of the 2018 Conference on Empirical Methods in Natural Language Processing.

Reasons for the score:
I think there are some outstanding questions regarding this paper that can be addressed in the author response, but until then, the proposed approach and associated results are less than convincing. I’m more than willing to increase my score following the author response.

---

> ### Author Response · Authors · 2020-11-17
> **Thanks. Evaluation mainly focusing on benchmark datasets for reading comprehension tasks which is problem we try to solve**
>
> 1."What happens when less than 20000 NewsQA examples are sampled for labeling is also not shown so it is hard to meaningfully interpret the results."
>
> The warm-start samples size for NewsQA is 10000, the difference between the compared algorithm below 20000 is not noticeable.
>
> 2."It is unclear whether this method generalizes to other models or not. Common and effective Active Learning baselines such as Query by Committee"
>
> Thanks for your suggestion. The "Query by committee" requires drawing two random hyperplanes which is not feasible to implement for the framework used here.
>
> 3.“ It would help if the authors show how the distribution of the metric of informativeness over examples, and as more and more data is labeled.“
>
> The sampling distribution of the samples is analyzed next to Equation (4). Based on our probability, the most uncertain sample will be selected with the highest probability.
>
> 4.“Are the results averaged over multiple runs?”
>
> As introduced in “Other Settings” page 6, the BERT-base is used as the pretrained model and fine-tuned for 2 epochs. The reported results are the performance of model trained after 2 epochs in each iteration recommended by https://github.com/huggingface/transformers/tree/master/examples/question-answering
>
> 5. “Why do you think this gap stays more or less the same with your method regardless of the amount of labeled data?”
>
> The main reason is that we use the adaptive threshold to control the sampling region. With available of more data, the model is improved and becomes less uncertain about the prediction results. In such case, random sampling has the advantage of diverse sampling while the ability of detecting uncertain samples of other algorithms is weaken.
>
> 6.“Typos ” “missing reference”
>
> Sorry for the typos. We will revise them and include the related reference in the revised version.

---

### Official Review · AnonReviewer3 · 2020-10-28
**Study has unclear novelty, and has some potentially important experimental flaws**

**Rating:** 3
**Confidence:** 3

**Review:**

### Summary

This paper proposes to apply uncertainty-based measures to guide the collection of training samples for reading comprehension. The paper describes a relatively simple metric to estimate model uncertainty of unlabeled examples, and develops an algorithm to sample examples that exhibit least model certainty. They describe a learning and regularization model for this scenario and evaluate their proposal on SQuAD and NewsQA datasets.

### Strong and weak points

Strong points:
- The use of uncertainty-based measures for active learning seem intuitive and elegant,
- The experimental evaluation contains a few interesting baselines and provide context to this work

### Weak points:

- The idea of using model uncertainty for sample efficiency is somewhat standard now. A few missing references on this are, for example [1], [2] and [3]. It is even described in blog articles online [4].  After reading the paper, it was not clear what part of the proposal is really novel, and which parts are well established in the literature.
Part of the work described seems to be dedicated to performing learning while sampling “in a loop” (Algorithm 1). Unless I’ve misunderstood the work, this assumption seems to be quite different than the premise of the work. After running an iteration of model training, new samples may be requested for annotation. In a real application, it would take hours, or most likely days, to receive new examples with labels. As such, I’m not sure it makes sense to design an algorithm that expects the new samples on every batch/iteration.  Given the time budgets, I would expect that models can be easily trained “from scratch” after new examples are available.

- I found the paper hard to follow at times. For example:
    - In Section 2.1.1 there seems to be use of informal notation: “We use a zero-one vector a to indicate the correct answer, ”. But at this point in the paper it’s unclear what the vector space represents (tokens in the input?). Later, in Section 3, a span-based representation is introduced.
    - The thresholding logic was hard to follow and understand the rationale. Are you simply sampling amongst the most uncertain samples? Can you not specify a $k$ to sample from? Why have an adaptive threshold?
    - The baselines models are not really explained clearly, with a single sentence per model,

- Overall, the results are quite inconclusive. Although there is modest improvement in SQuAD benchmark, it does not seem to be the case for NewsQA (the NewsQA results are only available as a chart, which makes comparison difficult). Also, in the SQuAD setup, the Random sampling method seems to generally be as good or better than the other baselines. There are no explanations as to why this may happen. Are the results being averaged over several runs? Do they represent the performance of a single run? Could the variations be attributed to natural variation in model training?

References used above:

[1] Heterogeneous uncertainty sampling for supervised learning
David D. Lewis and Jason Catlett
https://scholar.google.com/scholar?cluster=9211137857521772693&hl=en&as_sdt=0,33

[2] Deep Bayesian Active Learning for Multiple Correct Outputs
https://arxiv.org/abs/1912.01119

[3] You Need Only Uncertain Answers: Data Efficient Multilingual Question Answering
http://www.gatsby.ucl.ac.uk/~balaji/udl2020/accepted-papers/UDL2020-paper-113.pdf

[4] Active learning — Uncertainty Sampling (P3)
https://medium.com/@duyanhnguyen_38925/active-learning-uncertainty-sampling-p3-edd1f5a655ac

### Recommendation

I would recommend not to publish this work at ICLR at this time.  As described above, it is unclear what the novelty of this study is, besides applying an established technique (uncertainty-based sampling) to machine comprehension tasks. I have concerns regarding the premise of the learning setup (incremental learning assuming new examples can be sampled). Finally, the results seem inconclusive to me, and it is unclear whether statistical variations are dominating effects in SQuAD and NewsQA.

### Questions for authors

Please refer to the “weak points” described above.

### Additional feedback

- Besides the technical work, I think the paper could benefit from improved writing. Currently, some parts are a bit hard to read.
- Please separate the notation, task definition and model design into separate (sub)sections, so readers can read the paper linearly.
- Overall, the writing could use some editorial help.
- Page 5, the text describes the best performance for 5k examples as “64.07%”, however,  Table 1 states it is “64.03%”.

---

> ### Author Response · Authors · 2020-11-17
> **thanks. The main novel contribution is the novel uncertainty sample method for deep learning based reading comprehension task**
>
> 1.“clear what part of the proposal is really novel,”
> As stated in Section 1.1, we consider the problem of learning an MRC model in the label-demanding context, and we propose a novel algorithm that interleaves data annotation and model updating. In particular, there are two core components for this end: an unsupervised uncertainty-based sampling scheme that only queries labels of the most informative instances with respect to the currently learned model, and an adaptive loss minimization paradigm that simultaneously fits the data and controls the degree of model updating. Moreover, our approach is modular in nature, meaning that the community would benefit from this work by leveraging our techniques into more real-world problems (e.g. image classification) where the availability of labels is a major concern.
>
> None of the mentioned contributions are considered in [1,2,3].
>
> 2. “I’m not sure it makes sense to design an algorithm that expects the new samples on every batch/iteration.”
> The pipeline works in two ways. We can select a batch of labeled data by active learning algorithm proposed in this work instead of random sampling, the best performance could be achieved with much fewer number of labeled data. In the human in the loop annotation situation, it is feasible to select a subset from the pool of unlabeled data for annotation first which tends to improve the performance of model greatly. The size of subset can be customized.
>
> 3."models can be easily trained “from scratch” after new examples are available"
> There is the tradeoff between accuracy and latency. According to our empirical results, training from scratch takes much more time than training incrementally. For example, the Squad v1.1 dataset with batch size 1,000, training from scratch takes 58 hours while the incremental training takes 5 hours on one GPU.
>
> 4. “ at this point in the paper it’s unclear what the vector space represents”
> $a$ is the probability of best and second best answer candidates which is explained in Section 2.1.1. Intuition. The span representation is converted to $a$ which is clarified in Section 3 after Equation (6).
>
> 5."Are you simply sampling amongst the most uncertain samples? Can you not specify a k to sample from? Why have an adaptive threshold?"
> We select sample by probability defined in Equation (3) for samples in the region defined in Equation (2). We do not use $k$ in the algorithm. The threshold $\tau_t$ determines the uncertainty region. It is changed over iterations because the model is improved with available of more labeled data and becomes less uncertain about its prediction results.
>
> 6."the baselines models are not really explained clearly, with a single sentence per model"
> The baseline model follows the most popular BERT-based model which is fine-tuned for reading comprehension task. We have tried our best to explain the model in Section 3 under the page limit.
>
> 7."the Random sampling method seems to generally be as good or better than the other baselines."
> In terms of exact match, active learning algorithms outperform the random sampling method. Only when there are enough training samples, the model is already well trained, the random sampling outperforms several active learning algorithms.

---

### Official Review · AnonReviewer1 · 2020-10-28
**Valuable study on active learning applied to machine reading with limited novely**

**Rating:** 4
**Confidence:** 5

**Review:**

The paper is tackling the problem of labeling cost in Machine Reading comprehension.
The paper uses an uncertainty-based approach in the context of extractive machine reading to choose the point to annotate.
In addition, the paper uses an adaptive loss minimization schema that consists of penalizing large moves in the parameter space which strongly related trusted region types of approaches[1].
The authors compare to 6 baselines with mitigated results.
While the paper is well-written with reasonable experiments on two machine reading datasets, I can mention three limitations for acceptance which are

(1) the lack of strong novelty beyond uncertainty-based active learning applied to MRC: In its current status, the paper is proposing to measure uncertainty as entropy over the probability distribution of each token to be the start and end token of the contiguous string constituting the answer. I find this proposition pretty interesting but trivial compare to uncertainty based active learning literature applied to text [2, 3].

(2) lack of significant improvement on the considered datasets: First, the results are difficult to compare to SoA results (https://rajpurkar.github.io/SQuAD-explorer/) as the amount of data-point is different (Table 1). Second, the difference between the baseline methods and the proposed approach doesn't seem significant (Figure 4). Furthermore, it is currently not possible to precisely assess the result significance as the variance of the results isn't reported.

(3) lack of numeral results justifying the use of the adaptive loss in this context: unfortunately, the authors proposed a second contribution related to the loss function (section 2.2) which is not evaluated with an ablation study to assess its possible utility.

Refs
[1] Trust Region Policy Optimization, Schulman and al, 2015
[2] Uncertainty-based active learning with instability estimation for text classification, Zhu and al, 2017
[3] Active Learning Using Uncertainty Information, Yang and al, 2017

---

> ### Author Response · Authors · 2020-11-17
> **Thanks for the reviews. This paper proposes a novel uncertainty based sampling method**
>
> (1). ``the paper uses an adaptive loss minimization schema that consists of penalizing large moves in the parameter space which strongly related trusted region types of approaches[1].''
>
> Thanks for pointing out the interesting work [1]. The techniques look similar from high level, and it is a pleasure to know that the idea has been explored in reinforcement learning. However, we are more comfortable to accredit the originality of such localization scheme to [Balcan et al 2009] as we did in the paper, in view of the strong connection to active learning and the fact that the work [Balcan et al 2009] was published many years ahead of [1].
>
> (2) “ trivial compare to uncertainty based active learning literature applied to text [2, 3].”
>
> While [2,3] are titled with ``Uncertainty-based active learning'', both the uncertainty measure and the way of being active are significantly different from our work.
>
> Our framework is based on a neural network for reading comprehension. We first define the uncertainty score for samples Equation (1), the sampling region Equation (2) and select samples by probability in Equation (3). The whole pipeline is novel and we provide analysis that with our sampling strategy, the most uncertainty samples will be selected with highest probability as shown in Equation (4). To control the update of model, we employ regularizer between the new model and previously trained model as shown in Equation (5).
>
> [2] utilizes SVM which treats samples close to the hyperplane as uncertain samples as used in [Balcan et al. 2009]. [3] uses the probability produced by any classifier and solves the binary classification problem. Compared to the more involved algorithm in the current work, the techniques in [2,3] seem superficial and they may not compete with our BERT-based active learning algorithm.
>
> (3) “the results are difficult to compare to SoA results as the amount of data-point is different (Table 1). ”
>
> The Squad v1.1 is used in this work and the number of samples is consistent with the original work of [Rajpurkar et al., 2016]. As we highlighted in the paper, the main contribution of this work is to achieve competitive or better performance with much fewer number of samples with a novel technique of active learning, rather than engineering a new neural network for better performance.
>
> In this sense, we have demonstrate the effectiveness of our active learning algorithm on the BERT-base model on various datasets. Given the fixed parameters for all the other settings, our framework achieves 2~3% improvement with the same amount of the samples. Our framework also achieves the performance with 25% fewer labeled data which is significant as the data annotation for reading comprehension is known to be extremely expensive and time-consuming.
>
> (4) lack of numeral results justifying the use of the adaptive loss in this context
>
> We note that in margin-based active learning, the adaptive loss is always tied to the localized sampling; otherwise the reduction of label complexity is not guaranteed.

---

### Official Review · AnonReviewer2 · 2020-11-03
**A simulated application of active learning to SQuAD and NewsQA**

**Rating:** 5
**Confidence:** 4

**Review:**

Summary:
* This paper proposes a learning algorithm that interleaves data annotation and model updating to reduce the amount of labeled data needed.
* The algorithm requests labels for the most “informative” (or uncertain) instances with respect to the current model.
* The algorithm additionally proposes a loss that combines fitting data and controlling the degree of model update.
* Results are reported on SQuAD and NewsQA and the finding is that one needs 25% less samples for same or better performance.

Strengths:
* The work presents an interesting comparison of active learning algorithms on SQuAD and NewsQA. The results on SQuAD are promising, even though they are well below SOTA. It might have been more interesting to see this approach in action on a much more low resource dataset, perhaps publishing new labeled data to be used for the task.

Weaknesses:
* Novelty is limited. This paper seems to be a fairly straightforward application of active learning.
* Results are positive, but not exciting or surprising. Having to collect 25% less data is not so desirable, especially if there is suspicion that the active learning algorithm might introduce unwanted biases.
* On page 2, the authors discuss how their algorithm allows the model to only fit the new uncertain instances rather than retraining the model from scratch. However this improvement could be an unnecessary complication. If the focus of the work is on reducing the need for labeled data, wouldn’t it be better to simply retrain the model on all the labeled data every time a new annotated batch of data comes in?
* Even though the authors experimented only on Machine Reading Comprehension, there is no modeling specific to this task. This is good for generality, but it calls into question the choice of the authors to make MRC a central topic of the paper. It would have maybe been more convincing to frame this work as focused on active learning and then investigate its effect in multiple applications.
* MRC will often have multiple candidate answers with high probability, but these candidates are not truly different. They will in many cases correspond to mostly overlapping spans, or to similar answers appearing in different parts of the passage. Intuitively it seems that this algorithm would focus annotators on manually labeling these cases, which could arguably be suboptimal use of annotator time.
* No real analysis of the results is presented. It might have been interesting to see what data is actually being selected by the active learning algorithm or what questions the model was learning to answer as active learning progressed.

---

> ### Author Response · Authors · 2020-11-16
> **Thanks for comments, focusing on MRC as the annotation is expensive which makes 25% less required labeled data significant**
>
> Thank you very much for acknowledging our contributions and valuable feedback. Here are our responses in the hope to resolve your concerns.
> 1. "Having to collect 25% less data is not so desirable"
> As we mentioned in the introduction, the annotation for reading comprehension task is time consuming and expensive. The annotators are required to come up with questions and answers for each passage. It costs a median size funding for a faculty and several months to annotate SQUAD dataset, not to mention the quality validation process in the following.
> 2. "wouldn’t it be better to simply retrain the model on all the labeled data every time a new annotated batch of data comes in"
> That is an option, but takes much longer time than our current pipeline. Retraining the model with current available labeled data each time a batch of labeled data arrives takes much more time than training the model incrementally.
> 3. "It would have maybe been more convincing to frame this work as focused on active learning and then investigate its effect in multiple applications."
> We really appreciate the suggestion which is also mentioned in our future works. We focus on machine reading comprehension as it is a difficult/popular task and requires a large scale of annotated data. The active learning algorithm which significantly reduces the necessary annotated data is in high demand for MRC task.
> 4. "these candidates are not truly different. this algorithm would focus annotators on manually labeling these cases, which could arguably be suboptimal use of annotator time."
> Not sure if we understand this concern correctly. Our algorithm detects uncertainty questions based on the probability difference of the best candidate answer and second best candidate answer returned by currently trained MRC model. The active learning is performed on the unlabeled data, instead of annotated data.
> 5. "It might have been interesting to see what data is actually being selected by the active learning algorithm or what questions the model was learning to answer as active learning progressed."
> Thanks for the suggestion. We will include examples in the revised version.

---

### Decision · Program_Chairs · 2021-01-07
**Final Decision**

**Decision:**

Reject

**Comment:**

All reviewers agree that the current approach is very similar to traditional uncertainty-based active learning, and that the empirical results are inconclusive, so at this point the paper is not ready for publication.